# Willow Aboveground and Belowground Traits Can Predict Phytoremediation Services

**DOI:** 10.3390/plants10091824

**Published:** 2021-09-02

**Authors:** Béatrice Gervais-Bergeron, Pierre-Luc Chagnon, Michel Labrecque

**Affiliations:** Institut de Recherche en Biologie Végétale, Université de Montréal, 4101 Sherbrooke East, Montréal, QC H1X 2B2, Canada; pierre-luc.chagnon@umontreal.ca (P.-L.C.); michel.labrecque@umontreal.ca (M.L.)

**Keywords:** bioconcentration factor, brownfield, functional traits, phytoextraction, phytostabilization, Salix, short-rotation coppice, soil contamination, species diversity and trace elements

## Abstract

The increasing number of contaminated sites worldwide calls for sustainable remediation, such as phytoremediation, in which plants are used to decontaminate soils. We hypothesized that better anchoring phytoremediation in plant ecophysiology has the potential to drastically improve its predictability. In this study, we explored how the community composition, diversity and coppicing of willow plantations, influenced phytoremediation services in a four-year field trial. We also evaluated how community-level plant functional traits might be used as predictors of phytoremediation services, which would be a promising avenue for plant selection in phytoremediation. We found no consistent impact of neither willow diversity nor coppicing on phytoremediation services directly. These services were rather explained by willow traits related to resource economics and management strategy along the plant “fast–slow” continuum. We also found greater belowground investments to promote plant bioconcentration and soil decontamination. These traits–services correlations were consistent for several trace elements investigated, suggesting high generalizability among contaminants. Overall, our study provides evidence, even using a short taxonomic (and thus functional) plant gradient, that traits can be used as predictors for phytoremediation efficiency for a broad variety of contaminants. This suggests that a trait-based approach has great potential to develop predictive plant selection strategies in phytoremediation trials, through a better rooting of applied sciences in fundamental plant ecophysiology.

## 1. Introduction

Soil contamination represents a risk for ecosystems and human health [1]. Worldwide urbanization and industrialization are rapidly increasing the number of contaminated brownfields, which would benefit from sustainable remediation approaches [2]. Conventional remediation can be expensive and logistically challenging [3], while in situ phytoremediation offers economic, social and environmental benefits [1]. Yet, given the wide variation in plants’ abilities to tolerate, accumulate, immobilize or degrade contaminants [4], one of the main challenges of phytoremediation remains identifying and predicting the best candidates or combinations of plants to perform key remediation services.

Willows (*Salix* spp.) are often used for brownfield remediation because of their efficiency and versatility, as well as their capacity to establish on sites with harsh conditions [5]. They accumulate trace elements (TE), grow quickly, produce large amounts of biomass, tolerate a wide variety of stressors and resprout quickly after copping (aboveground biomass harvest) [6,7]. The genus *Salix* (Salicaceae) includes about 500 species worldwide and several hundreds of cultivars [8]. *Salix* species can bear distinct traits [9] and be associated with different microorganisms [10], which in turn cause variation in the phytoremediation services they can provide [11,12]. Using a diversity of willow species or cultivars may thus lead to the delivery of complementary services, especially under mixed contamination, as is often the case on brownfields [13,14].

To screen for species that promote phytoremediation services more efficiently under different environmental conditions, predictive frameworks linking plant identity and diversity to services must be developed. One such framework proposes that functional traits be used as proxies to predict ecological services [15,16,17]. Research on the relationship between biodiversity and ecosystem functioning has shown that plant trait diversity is likely a better predictor of key ecosystem functions than taxonomic diversity [18,19]. Moreover, trait-base approach can be predictively developed to maximize specific ecosystem services of plant assemblages [20]. Such an approach grounded in theory would free practitioners from the daunting cost- and labor-intensive task of screening species to identify the best candidates for field implementation [3]. Prior research using traits to guide species selection in phytoremediation has focused on proximal traits directly related to TE concentrations in aboveground tissues [21,22,23]. This cannot be applied to a large pool of species and ecosystems, because tissue concentration of a specific contaminant is a trait that (1) may be plastic and mirror soil conditions [22] and (2) is known for only a few species, thus restricting the pool of candidate species. A more effective approach would involve determining correlations between phytoremediation services and widely measured plant traits, such as those related to plant acquisition and conservation of resources [24,25]. For example, although Audet and Charest [22] were able to document tissue TE concentrations in 50 species in a meta-analysis, vast plant traits databases remain untapped in phytoremediation. For example, the international TRY Plant traits database holds information on specific leaf area (SLA) for >16,000 species, on leaf nitrogen content (LNC) for >12,000 species, and on more than 80 *Salix* species [26]. These willows even show distinct functional traits and follow the typical leaf economic spectrum (Appendix A). However, for these widely measured traits to be useful in phytoremediation, we need to demonstrate that they indeed covary with key phytoremediation services.

Traits vary among species, but also according to environmental filters, like disturbances [27]. In phytoremediation, willow short-rotation coppicing (SRC) (every 3 to 5 years) implies frequent and severe disturbances [6]. Because coppicing is likely to influence willow traits, with cascading impacts on phytoremediation services, it is important to understand how this practice will influence plant traits of interest.

Current phytoremediation approaches often lack predictability, partly because of our poor knowledge of which plant traits are associated with decontamination efficiency. Better anchoring phytoremediation in plant ecophysiology has the potential to drastically improve plant selection and phytoremediation success. In this study, we investigated the effect of willow diversity (plantations of one and four cultivars) and coppicing treatment on phytoremediation service delivery and plot-level plant traits. We also explored potential correlations between plant traits and phytoremediation services. We hypothesized that (1) willow diversity and coppicing treatment will increase phytoremediation services, (2) will modify community-level traits and (3) that these services and traits will be correlated together. This is a first step toward determining whether plant traits can be used as proxies to guide species selection in phytoremediation projects.

## 2. Results

### 2.1. Phytoremediation Services

According to linear mixed models (LMMs), diversity and coppicing treatments had varying effects on phytoremediation services (Figure 1), with willow polycultures plots showing better phytostabilization (Figure 1b, *p* < 0.05) but at the same time slightly lower root bioconcentration factor (BCF) (Figure 1d, *p* = 0.051), indicating root biomass overyielding. Coppicing, on the other hand, resulted in reduced phytoextraction (Figure 1a, *p* < 0.01), as expected given the aboveground biomass removal implied in coppicing, but also in reduced phytostabilization (Figure 1b, *p* < 0.01). Soil decontamination was not linked to either treatment, although many of the planted plots showed higher decontamination than the unplanted controls (Figure 1f). Soil physico-chemical properties also influenced some phytoremediation services. Organic matter (OM) content reduced phytostabilization (Figure 1b, *p* < 0.05) and bioconcentration in both total and root biomass (Figure 1d,e, *p* < 0.05), while initial TE concentrations were positively associated with translocation factors (Figure 1c, *p* < 0.01).

Cd and Zn were the most absorbed and accumulated TE in willow biomass. They had the highest total BCF (respectively 1.74 and 1.90), root BCF (respectively 0.49 and 0.80), and translocation factors (respectively 2.60 and 1.39) (Appendix A). Cu followed with higher root accumulation (BCF = 0.51) but lower translocation factor (TF) (TF = 0.31) (Appendix A). Counterintuitively, high Cd, Zn and Cu accumulation in willow biomass did not translate into more efficient soil decontamination: concentrations of these TE in soil increased between 2016 and 2019, by 9% for Cd, 65% for Zn and 31% for Cu (Appendix A).

### 2.2. Functional Traits

According to LMMs, coppicing influenced many plant traits typically associated with nutrient acquisition strategies. Coppiced plots showed lower leaf dry matter content (LDMC), root carbon content (RCC) and leaf pH, with higher SLA, leaf area (LA) and root nitrogen content (RNC) (Figure 2). Willow diversity, on the other hand, had only modest effects on community-level traits. *S. miyabeana* ‘SX67’ monocultures had higher root N and C content and leaf pH (Figure 2). Traits were also correlated with soil covariates: OM content increased leaf C and N content (LCC and LNC); clay content increased leaf pH and RCC (Figure 2, *p* < 0.05); and sand content reduced root dry matter content (RDMC) (Figure 2g, *p* < 0.1). Soils with higher TE concentrations tended to have lower leaf carbon content (LCC) (Figure 2f, *p* < 0.05) and marginally higher LA (Figure 2a, *p* = 0.05) and SLA (Figure 2b, *p* = 0.07).

### 2.3. Phytoremediation Services and Functional Traits

The redundancy analysis revealed interesting linkages between traits and phytoremediation services (Figure 3, R^2^ = 0.25, F = 1.56, *p* < 0.05). Soil decontamination was positively associated with RNC and RCC but negatively associated with LCC and specific stem density (SSD). Phytoextraction was mainly correlated with RCC and leaf pH, while phytostabilization was more efficient with plants having denser tissues (LDMC and RDMC), and lower SLA, LNC and RNC (Figure 3). For each phytoremediation service, different TE tended to cluster in the RDA biplot, indicating that correlations between willow traits and services were consistent regardless of the contaminant (Figure 3). The only outlier was TF service, as Pb, Mn and Cu translocation were more efficient with plants showing high LCC and SSD values, while Se, Cd and Zn translocation were correlated with RNC and RCC instead (Appendix A).

## 3. Discussion

This brownfield experiment proves that there are links between plant traits and phytoremediation. While individual services were inconsistently influenced by either willow diversity and coppicing (Figure 1), a canonical ordination revealed significant correlations between willow traits and phytoremediation, which was consistent across several contaminants. This suggests that phytoremediation may be more predictable if anchored in theory and plant ecophysiology.

### 3.1. Phytoremediation Services

Willow diversity had only modest impacts on phytoremediation services. It significantly improved phytostabilization (Figure 1b) while reducing root BCF (Figure 1d). This is explained by root biomass overyielding in willow polycultures, which is common in plant diversity studies [28]. It also supports other studies suggesting that niche partitioning among complementary species may promote willow biomass production [29], and help to cope with co-contamination and spatial heterogeneity [5,13]. Still, integrating plant diversity in phytoremediation projects remains a challenge, considering it often depends on species identity and characteristics [29,30]. However, the traits measured on the three willow species used here were similar to the ones registered on the TRY database (Appendix A) [26]. This suggests that even in stressful environments, such as a contaminated brownfield, the mean traits of willow do not drastically change from the mean database traits, highlighting the potential to use database traits for predictive approaches in phytoremediation.

Coppicing is expected to stimulate willow growth, especially for stems [6] and fine roots [31]. Such massive biomass removal and subsequent resprouts rejuvenate plant tissues, with traits expected to bear greater resemblance to those of younger plants. In line with such expectations, we found coppiced plants to have higher LA and SLA, and lower LDMC, which mirrors trait differences found between young and old trees [25,32]. Similarly, coppiced willows had higher RNC, indicative of younger root tissues with higher growth rates and faster nutrient acquisition [33]. However, coppicing reduced root biomass, as it reduced phytostabilization without altering TE bioconcentration levels (Figure 1). Longer-term studies would be needed to better appreciate the impact of root tissue rejuvenation through coppicing, as it may increase root mortality and turnover rate, as well as root:shoot ratio [31,34,35]. This could in turn have an impact on phytoremediation services by altering soil properties and, concurrently, TE mobility and bioavailability.

The significant impact of soil properties, included as covariates in our models, on phytoremediation services and plant traits, reiterates the importance of taking spatial heterogeneity into account in field phytoremediation studies [36]. Small-scale environmental heterogeneity is a known source of plant trait variation [37,38]. Soil fertility in particular is known to be associated with the leaf traits associated with resource management strategies [25,39]. In line with this, we found leaf N and C to be positively affected by soil OM content, which is an important component of soil fertility that stimulates plant growth [40]. Yet, our findings show that OM also reduced bioconcentration and phytostabilization, which could be explained by the efficient TE adsorption on exchange sites of soil OM, some of which was added at the beginning of our experiment in the form of compost. Contrary to our expectations, soil texture had only limited effects on phytoremediation, despite the known role of clay phyllosilicates in providing negative TE exchange sites [41].

### 3.2. Traits Predict Services

Our experimental factors had a few and inconsistent effects on phytoremediation services. However, these factors and spatially heterogeneous soil properties did influence plant trait values, and these traits were significant predictors of phytoremediation services. Moreover, our canonical ordination revealed trait covariance patterns similar to those typically reported in plant ecophysiological studies, with a clear trade-off between resource acquisition (SLA, LNC, RNC) and conservation (LDMC, RDMC, RCC) [24,25,42]. Indeed, our findings show that willow plantations with more conservative resource strategies fostered phytostabilization and phytoextraction. These results agree with phytoremediation studies on willows that recommend species with a high stress-tolerance capacity and resistant root systems [7]. A lot of species found under similar conditions, such as serpentine soils (low nutrients and high TE contents), also show conservative resource strategies [43]. However, it also disagrees with the common recommendation to favor fast-growing species in order to maximize harvests of contaminated biomass [12,44]. This apparent contradiction might have arisen because of our choice of closely related species of *Salix*, which are all considered relatively fast-growing species [6]. It is, therefore, possible that when only fast-growing species are compared together, stress-tolerant traits might contribute additional benefits in the context of phytoremediation. Moreover, in our study, many traits on this spectrum (LA, SLA, LDMC, RNC and RCC) were further affected by tissue age (coppicing treatment (Figure 2)) and so, indirectly, by the quantity of biomass produced. As a result, the plant economic spectrum gives insight on plant growth strategies as well as plantation maturity to evaluate phytoextraction or phytostabilization services in SRC willows.

Not only were plant traits more consistent predictors of phytoremediation services than our experimental treatments, but different TE tended to be similarly correlated with plant traits. This represents first evidences for a trait-based approach as a tool to improve the predictability of phytoremediation projects. However, increasing our understanding of the mechanistic nature of specific linkages between traits and services will require further empirical investigation. For example, phytoextraction was highly correlated to leaf pH, which was found to reflect leaf chemical composition (lignin, cellulose, cation and anion content) [45,46]. Thus, leaf pH might be a good predictor for the quantity of metallic ions accumulated in aboveground tissues but still might varies across plant species or environmental gradients. Likewise, much remains to be learned regarding the discrepancy between the translocation of mobile TE (e.g., Cd, Zn), which was favored by roots with higher N content (i.e., fast-growing, acquisitive roots), and the translocation of less mobile contaminants (e.g., Cu, Pb), which was rather promoted by higher aboveground investment (Appendix A) [41]. The latter could suggest that the translocation of less mobile TE in plants might require greater resources in aboveground tissues, such as transport proteins or phytochelatins [4,47]. Finally, traits related to evapotranspiration are an important aspect of plant ecophysiology that will be important to investigate in future studies on short-rotation coppice. These traits are likely to be a crucial driver of TE movement in the soil, with sometimes counterintuitive accumulation in the surface soil surrounding the plants, due to mass movements of TE with abundant transpiration in humid climates [48].

### 3.3. Limits

Here, we evidenced correlations between plant traits and phytoremediation services through a canonical ordination. The statistical power of this analysis may have been inflated by the simultaneous investigation of several TE in the same study (which may not fully comply with the row independence assumption of the analysis). Future studies could address this limitation by focusing on a single TE, while including more species from various clades of the plant phylogeny, or plants growing under contrasted climates. Moreover, soil TE concentrations may have been affected by different factors, such as the addition of a 10 cm compost layer on the topsoil or the high spatial heterogeneity, inherent to real-field contaminations [36]. In addition, the observed increases in final soil TE (ex. Cu and Zn) can be explained by the strong evapotranspiration of willow, acting as a soil solution pump [49], and converging contaminants toward their root systems and the surrounding soil horizons [48,50]. However, the present study does not provide mechanisms for the phytoremediation patterns but demonstrates the interest in conducting additional comparative ecological studies to correlate plant traits with their ability to provide phytoremediation services.

## 4. Materials and Methods

### 4.1. Sites Description

The experiment was conducted on a brownfield under phytoremediation for four years. Located in a petrochemical sector of Montreal East (Quebec, Canada, 45.638 N, 73.511 W), the site was found to be moderately contaminated with polycyclic aromatic hydrocarbons (PAH) and TE, notably As, Ba, Cd, Cu, Mn, Pb, Se and Zn (Table 1) [51]. The clayed soil was slightly alkaline (pH = 7.6 ± 0.1), with 8.7% OM and 4.1 g/kg total nitrogen content determined by combustion [52]. The site was characterized by a humid continental climate, with average summer temperatures of 20.2 °C (annual mean temperature of 6.9 °C) and annual precipitation around 998 mm [53].

### 4.2. Experimental Design

The experiment was based on a full factorial design with two factors (i.e., willow diversity and coppicing) distributed randomly across four blocks, following up on an earlier experiment established in May 2016 [51]. Vegetation and debris were removed from the experimental plots and microcuttings of willows were planted according to the technique developed by Guidi Nissim and Labrecque [54]. Briefly, a 10 cm layer of compost (negligible TE, Table 1) was spread on plots (except the controls) and 120 microcuttings (5 cm long) per m^2^ were added to the compost. The same planting density was used for polycultures, implemented by mixing 30 microcuttings of each of the four selected cultivars (see below), per m^2^. Each block included three 4.5 m × 5 m plots: (1) a non-vegetated control, (2) a monoculture of *Salix miyabeana* ‘SX67’ and (3) a polyculture including four willow cultivars: *Salix miyabeana* ‘SX67’, *Salix miyabeana* ‘SX61’, *Salix purpurea* ’Fish Creek’ and *Salix gmelinii* ’India’. Willow varieties were selected based on their commercial availability but also to ensure a diversity of growth patterns as well as leaf morphologies. *S. miyabeana* cultivars are fast growers with high SLA, while *S. gmelinii* grows more slowly and *S. purpurea* has tougher leaves with high LDMC [51]. Before the beginning of the fourth growing season, in early May 2019, the western half of each plot was coppiced by cutting aboveground vegetation at 10 cm above the soil surface. By the end of the fourth growing season, in October 2019, the coppiced plantations were approximately 1.5 m high while non-coppiced plantations were around 4 m. This resulted in five combinations of treatments: willow diversity (0-1-4 cultivars) and coppicing (with or without). Plots were not fertilized, but were weeded once a month for the first year and watered as needed during seasonal dry periods.

### 4.3. Phytoremediation Services

To measure phytoremediation services, we selected six indices that evaluate different facets of phytoremediation, such as the quantity or ratio of TE accumulated in plant tissues [21,30]. (1) Phytoextraction and (2) phytostabilization were calculated, respectively, by multiplying aboveground and belowground biomass (t/ha) and their TE concentrations (kg/ha). (3) The TF (unitless) is the ratio between aboveground and belowground TE concentration and reflects the plant’s ability to accumulate more in aboveground than belowground tissues. (4) Total BCF or (5) root BCF (unitless) were calculated as the ratio between tissue TE concentration (shoots + roots, or roots only, respectively) and soil TE concentration. (6) Soil decontamination was calculated as the proportional difference in soil TE concentration between 2016 and 2019 (soil concentration of 2016 minus 2019 and divided by 2016).

At the end of the fourth growing season (October 2019), we harvested and pooled the aboveground biomass of three quadrats (25 cm × 25 cm) and extracted two soil cores (1.4 dm^3^ ± 0.6 into the 0–30 cm horizon) from each plot (controls, one and four cultivars and coppiced or not). The soil was thoroughly mixed (with roots removed) and air-dried at room temperature (22 °C) to measure OM content by loss on ignition and texture using the hydrometer method [55]. Willow aboveground and belowground biomass was dried (72 h at 60 °C) and weighed. To estimate TE concentrations of plant tissues and soils (2016 and 2019), we ground, sieved (0.5 mm) and digested plant biomass and soils in concentrated HNO_3_ at 120 °C for 5 h (protocol adapted from Wilson et al. [56]). We measured As, Ba, Cd, Cu, Mn, Pb, Se and Zn concentrations by inductively coupled plasma mass spectrometry (ICP-MS) (Perkin Elmer NexION 300).

### 4.4. Willow Trait Measurement

We measured ten traits (SLA, LA, LDMC, Leaf pH, LNC, LCC, RDMC, RNC, RCC and SSD) in willow composite tissue samples from every vegetated plot (representing community-level mean traits). These traits were selected based on their known association with ecological functions of interest in phytoremediation, such as tolerance to disturbance, nutrient acquisition or stress response (Table 2). According to the protocols of Pérez-Harguindeguy et al. [57], we measured eight traits (RDMC and SSD excluded) on the vegetation (coppiced and non-coppiced), while it was at its peak, in August 2019. Briefly, we collected fine roots from three pooled soil cores and healthy leaves from the upper 15 cm of all the stems present in three pooled 25 cm × 25 cm quadrats. We quantified LCC, LNC, RCC and RNC by dry combustion with an Elementar Vario MICRO cube [58]. We also measured, in Auguste 2019, the RDMC by calculating the mass ratio between dry and fresh fine root mass (48 h at 60 °C) [59]. Finally, at the end of the growing season, in October 2019, we measured SSD (dry mass per unit volume) by harvesting the bottom 15 cm of shoots comprised within three randomly placed 25 cm × 25 cm quadrats, and drying them at 70 °C for 72 h [57].

### 4.5. Statistical Analysis

We tested the impacts of willow diversity and coppicing on phytoremediation services through LMMs using the R package lme4 [72] (R studio, version 4.0.3 [73]). We built equivalent models for all six phytoremediation services outlined above. The response variable was an arithmetic mean of the eight TE (with phytoextraction and phytostabilization being first standardized (z-scores) to take into account the wide scales variations among TE). Soil properties (OM, clay and sand content and TE initial concentrations) were included as covariates, to take into account heterogeneity in soil conditions on the site. Block identity was included in the models as a random intercept. We verified the absence of any strong collinearity between explanatory variables (variance inflation factor < 5) [74] before model construction. Initial soil TE concentrations were not included in the model for soil decontamination, as they are included in the service calculation. Using similarly constructed LMMs, we modelled willow trait variation according to treatments, including soil properties as covariates. Two missing data (LNC and LCC from a single plot) were replaced by the variables means [75]. Additionally, the 95th percentile (capping transformation) replaced one disproportionately high outlier in RDMC.

We explored correlations between phytoremediation services and plant traits using redundancy analysis (RDA) in the R package vegan [76]. All services, each with eight corresponding TE, were standardized and included separately in the response *Y* matrix. We removed LA from the plant trait matrix due to its strong collinearity with LDMC and SLA (variance inflation factor > 5) [74]. We tested the significance of the RDA through a pseudo-*F* ratio test and estimated its adjusted coefficient of determination (R^2^).

## 5. Conclusions

This brownfield experiment provides the first evidence for a trait-based approach in phytoremediation as a tool to improve predictive power. While our treatments had only modest effects, we found consistent correlations between plant traits and phytoremediation services. More specifically, phytoextraction and phytostabilization of willows can be predicted by the plant economic spectrum, which represents growth strategies and plantation maturity. Higher root resources allocation (RNC and RCC) also predicted better bioconcentration factors and soil decontamination. Even more interestingly, these correlations were found independent from TE properties (except for translocation factor), suggesting that a plant trait approach could not only free phytoremediation predictions from taxonomic specificities but also contaminant ones. What remains to be explored is to what extent plant traits are free from any context dependencies and local site contingencies (e.g., climate and nature of soil contamination). At the very least, our results suggest that careful integration of plant traits in phytoremediation projects using short-rotation coppices willows could help to promote plantation success and predictability. Future work will expand the knowledge and boundaries of trait-based approaches in phytoremediation.

## Figures and Tables

**Figure 1 plants-10-01824-f001:**
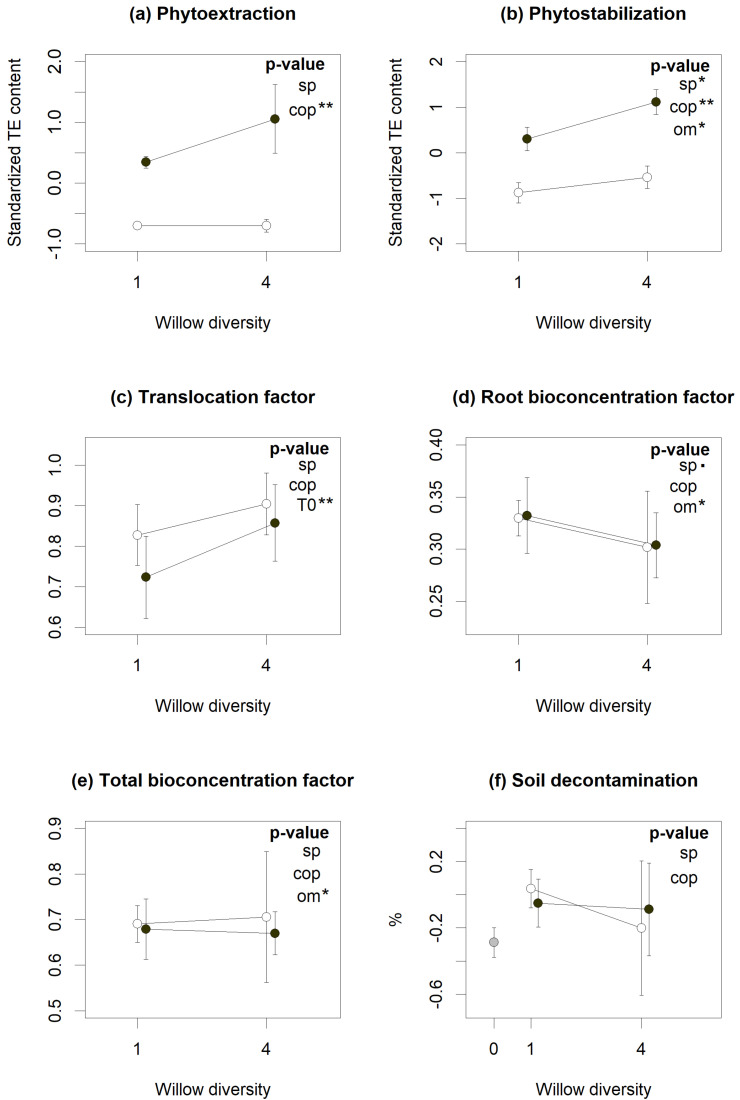
Influence of willow diversity (number of cultivars) (sp), coppicing (cop) and environmental factors (Organic matter (OM), clay (clay) and sand (sand) content and initial contamination content (t0)) on phytoremediation services (**a**−**f**). Dots represent the services means across the eight TE and the standard error bars of each cross-treatment. The unplanted control (grey symbol), coppiced (open symbols) and non-coppiced (closed symbols) treatments are aligned above the number of species unplanted (0), monoculture (1) or polyculture (4) plots). According to linear mixed models, treatments or factors with significant *p*-values are indicated by (.) for *p* < 0.10, (*) for *p* < 0.05 and (**) for *p* < 0.01.

**Figure 2 plants-10-01824-f002:**
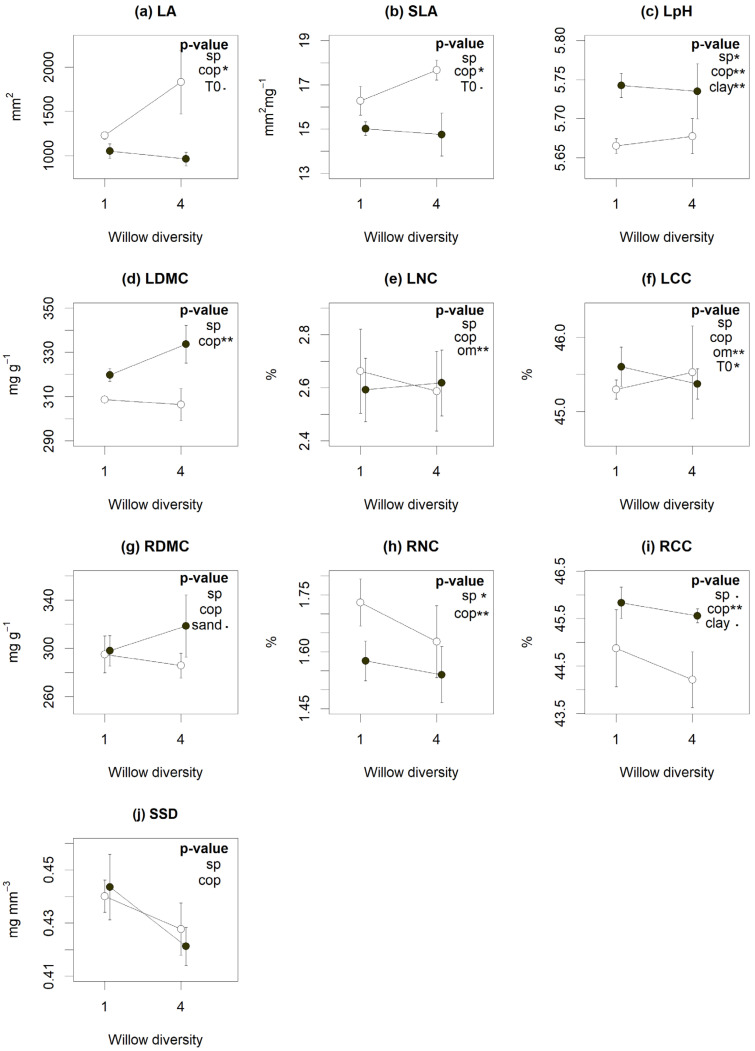
Influence of willow diversity (number of cultivars) (sp), coppicing (cop) and environmental factors (Organic matter (OM), clay (clay) and sand (sand) content and initial contamination content (t0)) on functional traits (**a**–**j**). Dots represent the traits means and bars the standard error of each cross-treatment. The coppiced (open symbols) and non-coppiced (closed symbols) treatments are aligned above the number of species (monoculture (1) or polyculture (4) plots). According to linear mixed models, treatments or factors with significant *p*-values are indicated by (.) for *p* < 0.10, (*) for *p* < 0.05 and (**) for *p* < 0.01. *Abbreviations*: leaf area (LA), specific leaf area (SLA), leaf pH (LpH), leaf and root dry matter content (LDMC and RDMC), leaf nitrogen and carbon content (LNC and LCC), root nitrogen and carbon content (RNC and RCC) and specific stem density (SSD).

**Figure 3 plants-10-01824-f003:**
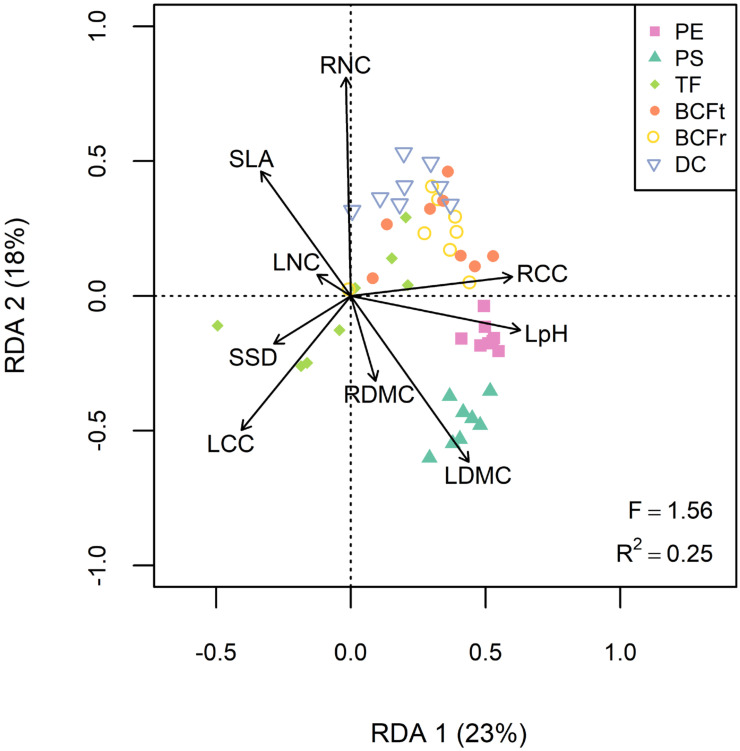
Redundancy analysis (RDA) regressing the six phytoremediation services for every TE (As, Ba, Cd, Cu, Mn, Pb, Se and Zn) to the functional traits. *Abbreviations:* Phytoextraction (PE), phytostabilization (PS), translocation factor (TF), total bioconcentration factor (BCFt), root bioconcentration factor (BCFr), soil decontamination (DC), leaf area (LA), specific leaf area (SLA), leaf pH (LpH), leaf and root dry matter content (LDMC and RDMC), leaf nitrogen and carbon content (LNC and LCC), root nitrogen and carbon content (RNC and RCC) and specific stem density (SSD).

**Table 1 plants-10-01824-t001:** TE concentrations (mg per kg of soil), soil characteristics and compost characteristics were measured at the beginning of the experiment in 2016 [51]. Copper, selenium and lead had initial concentrations that fall within the first contamination class (A-B) of Quebec’s legislation.

TE	Soil TE Mean (mg/kg)	SD	Compost TE Mean (mg/kg)	SD	Soil Characteristics	Mean	SD	Compost Characteristics	Mean	SD
As	5.12	(2.09)	-	-	pH	7.6	(0.1)	pH	5.9	(0.1)
Ba	149.19	(34.21)	-	-	CEC (meq/100 g)	37.8	(7.9)	CEC (meq/100 g)	33.0	(0.3)
Cd	0.78	(0.38)	-	-	OM (%)	8.7	(2.8)	OM (%)	27.6	(1.2)
Cu	92.76	(52.69)	2.13 *	(0.09)	Total N (g/kg)	4.1	(1.5)	Total N (g/kg)	9.1	(0.8)
Mn	573.53	(187.53)	18.6 *	(2.3)	Clay (%)	42.5	(9.9)	P (kg/ha)	171 *	(0.1)
Pb	80.72	(46.09)	-	-	Silt (%)	38.3	(15.3)	K (kg/ha)	99 *	(0.1)
Se	1.88	(1.26)	-	-	Sand (%)	19.2	(13.3)	C: N ratio	19.9	(0.8)
Zn	73.69	(36.11)	11.0 *	(0.90)						

Abbreviations: Trace element (TE), cations exchange capacity (CEC) and organic matter (OM). * Mehlich III fraction.

**Table 2 plants-10-01824-t002:** Functional traits associated to plant strategies, ecological functions and environmental properties.

Functional Traits		Units	Functions and Strategies
SLA	Specific leaf area	mm^2^/mg	Growth rate, photosynthetic capacities [57,60,61], carbon investment, stress tolerance, nutrient acquisition strategy [57,62,63], leaf longevity [63,64], ruderal strategies [65] and soil fertility [66]
LA	Leaf area	mm^2^	Photosynthetic capacities [61], environmental responses and tolerance [57], light interception, leaf and plant size [63] and competitiveness [65]
LDMC	Leaf dry matter content	mg/g	Slow growth rate [57], litter decomposition, nutrient retention and acquisition strategy [62,67], stress tolerance strategies [65] and tolerance to disturbances [57,68]
LpH	Leaf pH		Nutrient content (cations) and tissues chemistry [57], leaf digestibility, litter decomposition and pH, leaf C:N ratio and leaf lignin and cellulose content [45,69]
LNC	Leaf nitrogen content	mg/g	Growth rate and litter decomposition [62], nutrient acquisition strategies, photosynthetic capacities, herbivory potential [63], soil fertility [66] and tolerance to disturbance [68]
LCC	Leaf carbon content	mg/g	Soil C [68], leaf structure investment, tolerance to disturbance [70], life form, lignin content and chemical composition [71] and leaf digestibility [45]
RDMC	Root dry matter content	mg/g	Tolerance to herbivory and stress, root decomposition rate [62], root growth rate and resource acquisition strategies [59]
RNC	Root nitrogen content	mg/g	Root respiration, root growth rate, root decomposition rate, root metabolic activity [62]
RCC	Root carbon content	mg/g	Life form, lignin content and chemical composition [71]
SSD	Specific stem density	mg/mm^3^	Hydraulic capacity [57], decomposition, defence capacities, resistance to stresses [62], growth rate, mortality risk [62,63], longevity [64] and carbon storage [62,64]

## Data Availability

The data presented in this study are openly available in the Dataverse repository of the University of Montreal at doi.org/10.5683/SP2/2ZJHKP.

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
