# Peer review of "Willow Aboveground and Belowground Traits Can Predict Phytoremediation Services"

_plants, 2021, doi:10.3390/plants10091824_

Round 1

Reviewer 1 Report

The manuscript entitled "Willow aboveground and belowground traits can predict phytoremediation services" explored how the community composition, diversity and coppicing of willow plantations, along with their effects on community-level plant functional traits, influenced phytoremediation services in a four-year field trial. The study provides evidence that a trait-based approach can help increase predictive power in phytoremediation, even for short taxonomic ranges and with different contaminants. It is an interesting paper and the subject certainly falls within the general scope of Plants journal. However, it is only recommended to be published after the following concerns being addressed and corrections being made.

  1. The Abstract is not a good writing, and it should clear the significance and the innovation of the study. In addition, it is better to list some exact data of the results.

  1. The section of Introduction was not well structured, and the latest literatures were cited inadequately. The authors should provide rational for why study was conducted and in what aspects it is novel from previously conducted similar studies. In addition, the scientific hypothesis need to be clear, and the logical relation of the research gap and this study was not well listed in the last paragraph of Background.

  1. In the section of Materials and Methods, some key information was missing in the description of experiment. Please check the whole section carefully and add those details. For example: How about the fertilization management? More details for TE concentrations of plant biomass and soils determination should be listed.

  1. The conclusion is too brief. Conclusion should summarize all the findings in this study, as well as the research meaning of the findings.

In summary, the MS has its merits, it could be considered for publication with minor revision.

Reviewer 2 Report

The paper contributes to increased predictability of phytoremediation of contaminated sites. It gives new scientific insights into phytoremediation which may improve its application in the future. It was an interesting read.

The paper addresses numerous willow traits, resulting in numerous abbreviations that are not always used consistently which can be challenging for the reader. Please check the use of abbreviations throughout the paper and supplementary material.

Abstract: remove the numbering of chapters in the abstract text.

Line 36: “spp.” not in italics

Line 67: Change »sppecies« to »species«

Results section: since the »Materials and methods« chapter comes at the end, there are some unexplained abbreviations in the »Results« chapter. Please explain them when they appear for the first time in the text (BCF in line 85, BCFr in line 98, LDMC and RCC in line 104, RNC in line 105, OM in line 107, RDMC in line 110, LCC in line 111).

Line 101: You mention that Cd, Zn, and Cu in the soil increased between 2016 and 2019. Could it be because of the addition of the compost?

Line 114: Please put “nb” in full word.

Figure 1 and Figure 2 could be designed in the same style, i.e. x-axis in Fig. 1 is “Willow diversity”, while in Fig. 2 it’s “Species number”; in Fig. 1 has titles of each graph in full words, while Fig. 2 has abbreviations (I suggest full words in both). If graphs have titles, then the y-axis can only have units.

Line 147: Change “proof” with “proves”.

Line 155: Change (Fig. 1, b) to (Fig. 1b) and (Fig. 1, d) to (Fig. 1d) as used in the Results chapter or vice versa.

Line 160: Change “depend” to “depends”.

Line 247: Change” the font size of the location

Line 254: Change »Table 2« to »Table 1«; explain the TE abbreviation as it appears for the first time in the Table/Figure section and correct »Kg« to »kg« in the table title. Corrections of the table:

  • Add a footnote to the table and explain the CEC abbreviation.
  • Correct »Kg« to »kg« when presenting P in compost.
  • Total N is given in mg/kg for soil and in g/kg for compost. Please express in the same units for better comparison.
  • Are P and K given as total P and K or as plant-available P and K? Please explain in the text.

Line 262: Was compost added also to the unplanted plots? Have you measured TE in the compost? Adding 10 cm of compost to 30 cm of examined soil core may significantly influence the TE concentrations and soil decontamination results. Please address this in the discussion.

Line 281: change “ton/ha” to “t/ha”

Lines 284-285: You are describing “root BCF” and “total BCF” in the Results chapter but here you are introducing “BCFr” and “BCFt”. Please unify throughout the paper.

Line 290: add space between the number and the unit (25 cm)

Line 291: add space between the number and the unit (0-30 cm)

Line 292: add space between the number and the unit (22 °C)

Line 294: add space between the number and the unit (72 h at 60 °C)

Line 296: add space between the number and the unit (120 °C for 5 h)

Line 304: You measured willow traits in August 2019 while growth was at its peak. Earlier in Line 272, you describe that half of each plot was coppiced in May 2019. Are the traits measured in August referring only to non-coppiced halves of the plots or both? Please explain this in the text. When does the growth season start – was there enough time for regrowth of coppiced willows?

Line 306: add space between the number and the unit (25 cm)

Line 312: add space between the number and the unit (48 h at 60 °C)

Line 315: add space between the number and the unit (70 °C for 72 h)

Reviewer 3 Report

Phytoremediation is one of hot-spot in environmental sciences.

Soil pollution by phytoremediation is one of very important issues
in environmental studies. It is very interesting and useful that the
 authors have investigated the effect of the community composition, diversity and coppicing of willow plantations, along with their effects on community-level plant functional traits, on phytoremediation services. The subject of the study was new and had novelty. In general, the MS was written well. The text is clear and easy to read. The conclusions was based on the results. Hence, It is recommend to be published in the present form. 
